# Prediction Models of Obstructive Sleep Apnea in Pregnancy: A Systematic Review and Meta-Analysis of Model Performance

**DOI:** 10.3390/diagnostics11061097

**Published:** 2021-06-15

**Authors:** Sukanya Siriyotha, Visasiri Tantrakul, Supada Plitphonganphim, Sasivimol Rattanasiri, Ammarin Thakkinstian

**Affiliations:** 1Department of Clinical Epidemiology and Biostatistics, Faculty of Medicine, Ramathibodi Hospital, Mahidol University, Bangkok 10400, Thailand; sukanya.sii@mahidol.edu (S.S.); supada.pli@mahidol.edu (S.P.); sasivimol.rat@mahidol.ac.th (S.R.); ammarin.tha@mahidol.edu (A.T.); 2Medicine Department, Division of Sleep Medicine, Ramathibodi Hospital, Mahidol University, Bangkok 10400, Thailand; 3Sleep Disorder Center, Ramathibodi Hospital, Mahidol University, Bangkok 10400, Thailand

**Keywords:** prediction model, gestational obstructive sleep apnea, systematic review and meta-analysis

## Abstract

Background: Gestational obstructive sleep apnea (OSA) is associated with adverse maternal and fetal outcomes. Timely diagnosis and treatment are crucial to improve pregnancy outcomes. Conventional OSA screening questionnaires are less accurate, and various prediction models have been studied specifically during pregnancy. Methods: A systematic review and meta-analysis were performed for multivariable prediction models of both development and validation involving diagnosis of OSA during pregnancy. Results: Of 1262 articles, only 6 studies (3713 participants) met the inclusion criteria and were included for review. All studies showed high risk of bias for the construct of models. The pooled C-statistics (95%CI) for development prediction models was 0.817 (0.783, 0850), *I*^2^ = 97.81 and 0.855 (0.822, 0.887), *I*^2^ = 98.06 for the first and second–third trimesters, respectively. Only multivariable apnea prediction (MVAP), and Facco models were externally validated with pooled C-statistics (95%CI) of 0.743 (0.688, 0.798), *I*^2^ = 95.84, and 0.791 (0.767, 0.815), *I*^2^ = 77.34, respectively. The most common predictors in the models were body mass index, age, and snoring, none included hypersomnolence. Conclusions: Prediction models for gestational OSA showed good performance during early and late trimesters. A high level of heterogeneity and few external validations were found indicating limitation for generalizability and the need for further studies.

## 1. Introduction

Obstructive sleep apnea (OSA) is a common disorder, characterized by repetitive upper airway collapse during sleep specifically apnea and/or hypopnea leading to oxygen desaturation, arousal, sleep fragmentation, sympathetic activation, and endothelial dysfunction [1,2,3]. Long-term cardiovascular consequences are shown both in men and women, despite the lower OSA prevalence in women [4]. OSA in women is often less perceived and underdiagnosed because of the “unclassical symptom” of OSA found in women [5]. In addition, snoring and witnessed apnea, the hallmark of OSA symptomatology were less reported in premenopausal women when compared to post-menopausal women [6,7]. Altogether, OSA in premenopausal women is underrecognized and may be problematic when these women become pregnant.

Gestational OSA had been shown to increase adverse maternal and fetal outcomes such as preeclampsia, gestational hypertension/diabetes, and preterm birth [8]. Prevalence of gestational OSA increased as increasing obesity in the population [9]. Despite evidence showing that early diagnosis and treatments of gestational OSA could improve pregnancy outcomes [10,11], diagnosis of gestational OSA is challenging given the difficulty in access to standard polysomnography (PSG) and together with unawareness of OSA in pregnancy. Therefore, there is a need for an accurate OSA screening tool specific to the pregnant population. However, conventional OSA screening questionnaire (i.e., Berlin questionnaire, Epworth sleepiness scale, ESS) had poor performance during pregnancy with pooled concordance statistics (C-statistics) of 0.68 (95% confidence interval, CI: 0.08, 0.98) [12]. Screening of OSA during pregnancy based on these questionnaires and/or self-reported symptoms alone may be less predictive as excessive daytime sleepiness (EDS), frequent nocturnal awakening, and fatigue are also commonly reported as physiological changes of pregnancy itself [12,13].

A prediction model is an equation constructed by using various statistical models to quantify the risk that an individual person would develop an event of interest [14]. Various prediction models specific for gestational OSA were developed and validated [15,16,17,18,19,20], as well as external validations of some general prediction models for OSA in pregnancy [13,19,21].

As for our knowledge, performance of the prediction models of gestational OSA has yet been systematically reviewed. This systematic review was conducted which aimed to summarize their performances (i.e., calibration and discrimination) and number of predictors that had been used in developed and validated phases for predicting OSA during pregnancy.

## 2. Materials and Methods

### 2.1. Search Strategy and Study Location

This study had been registered on the international prospective register of systematic reviews (PROSPERO) with registration number CRD42021237996. An extensive literature search was performed on 2 major databases in MEDLINE (from 1996 to 28 February 2021), and Scopus databases (from 1980 to 28 February 2021) as recommended by the PRISMA guideline [22]. Search terms were constructed according to the PICO principles (i.e., participants, intervention, comparator, and outcome): pregnancy (MeSH), “pregnant women”, parturient, gestation*, obstetric”; “sleep questionnaire”, screening, “prediction model”, predictors, “prediction tool”, “risk score”; “polysomnography”, PSG, sleep test, “home sleep test”, “Watch-PAT; and obstructive sleep apnea (MeSH), “obstructive sleep apnea’, “sleep apnea”, OSA, “sleep-disordered breathing”, snoring. The details on search terms and search strategies for each database are described in supporting information in Appendix A (Table A1 and Table A2).

### 2.2. Inclusion and Exclusion Criteria

Any type of observational study (cross-sectional, cohort, or case control) or randomized controlled trial published in any language was included in the review if it met all the following inclusion criteria: (1). studied in pregnant women, (2). developed or externally validated at least one multivariable model for predicting diagnosis of OSA during pregnancy, and (3). had outcome of interest as OSA diagnosed by objective sleep tests including PSG, or home sleep apnea test (HSAT).

Exclusion criteria for studies were any of the following; (1). They were reviews or case reports, (2). Had insufficient data for pooling despite several attempts to contact authors, or (3). were multiple publications of the same original study.

All identified articles were combined and duplicates were excluded. Studies were independently selected by reviewers (SS, SR) by screening titles and abstracts. If a decision could not be made based on abstracts, full articles were retrieved. Disagreements between reviewers were adjudicated by a third reviewer (VT).

### 2.3. Data Extraction

Two independent researchers (SS, VT) independently extracted the data. The general characteristics of studies including author, publication year, study design, number of subjects, study phases (i.e., development and internal/external validations), and diagnosis of OSA were extracted. If the prediction model was a development model, specific information about the model construct (i.e., type of statistical model, predictors and selections, creating scores using coefficients or their exponentials) were extracted. Model performance in discrimination by C-statistics along with 95% CI was extracted by study phases. In addition, model performance in calibration was also extracted. Details on the prediction models and operational definitions are described in Appendix B.

### 2.4. Reference Test

The outcome of interest was diagnosis of OSA during pregnancy based on objective sleep tests including PSG, or any type of HSAT including Watch-PAT^®^ (Itamar Medical, Isarael). The criteria for diagnosis was defined according to the original studies, i.e., either apnea-hypopnea index (AHI) or respiratory disturbance index (RDI) ≥ 5 events/hour.

### 2.5. Risk of Bias Assessment

We appraised the risk of bias (ROB) of the studies’ developing or externally validating prediction models using the Prediction Model Risk of Bias Assessment Tool (PROBAST) for systematic reviews [23,24]. This contains multiple signaling questions in four different domains: participants, predictors, outcome, and analysis. Signaling questions are answered as “yes”, “probably yes”, “probably no”, “no”, or “no information” where yes and no mean low and high risk of bias. Overall ROB is judged as low risk if all domains are considered low risk, high risk if at least one of the domains is considered high risk. Two researchers (SS, VT) independently assessed the ROB.

### 2.6. Statistical Analysis

We calculated and reported descriptive statistics to summarize the characteristics of the models. We calculated the median and interquartile range for continuous variables and the respective percentages for categorical variables. For the prediction models that were examined in more than 2 independent datasets, we applied a random effect meta-analysis to calculate the summary estimates of C-statistics and calibration separately by study phase. We followed a recently published framework for the meta-analysis of prediction models [23,24]. For those studies reported only C-statistics but not for dispersions (e.g., standard error (SE) or 95% confidence interval), their SEs were estimated following a formula. Heterogeneity was assessed using Cochrane Q test and its degree was quantified by the *I*^2^. All statistical analyses were performed using STATA version 16.1 (StataCorp^®^, College Station, TX, USA), with a significance threshold *p*-value < 0.05 (2-sided).

## 3. Results

### 3.1. Description of the Included Studies in the Systematic Review

A total of 1262 studies were identified but 6 studies (3713 participants) met our inclusion criteria and included in meta-analysis, (see Figure 1) [15,16,17,18,19,20]. The number of participants were largely driven by a study of a cohort of pregnant women (3264 participants) during second–third trimesters [17]. Characteristics of these studies were described, (see Table 1 and Table 2). All studies were prospective cohorts of pregnant women with a total of 29 prediction models [15,16,17,18,19,20]. Two studies involved high risk pregnancy, in which one study defined as chronic hypertension, pre-gestational diabetes, obesity, and/or history of preeclampsia [15]; while another study defined as extreme obesity (BMI ≥ 40 kg/m^2^) [18]. All of the studies included mixture of ethnicity [15,16,17,18,19,20], with majority of whites (range 20–60.4%), while 2 studies of the same dataset had 75% of African Americans [19,20]. One study included only nulliparous participants [17]. Two studies screened for OSA once, each during first and third trimesters [15,18]. Four longitudinal studies performed OSA screening twice consisted of 3 studies during first to second–third trimesters [17,19,20], and one study during second to third trimesters [16]. One of these studies reported new-onset of OSA [17]. Diagnosis of OSA was made by PSG in three studies [16,19,20], HSAT in two studies [17,18], and Watch-PAT in 1 study [15]. Criteria for diagnosis of OSA was mainly based on AHI ≥ 5 events/hour in 5 studies [15,17,18,19,20], except 1 study using RDI ≥ 5 events/hour [16]. All studies were development phases, in which only one study had performed internal validation [17], whereas three studies had externally validated previous models [16,19,20].

### 3.2. Risk of Bias Assessment

Results of PROBAST are described, (see Figure 2 and Table 3). The overall ROBs were high risk as for the outcome and analysis domains. For the outcome domain, 1 study had workup and spectrum biases by not including all participants recruited during second trimester, only small subset of participants with either end of OSA risk (high vs. low) were followed and underwent PSG during the third trimester [16]. Although PSG is the gold standard diagnostic test for OSA, only 3 studies performed PSG. Three other studies used HSAT with different criteria for hypopneas [15,17,18]. One study used Watch-PAT, a wrist-worn device using a peripheral arterial tonometry (PAT), finger plethysmography, and pulse oximeter [25]. Watch-PAT had been validated and shown good accuracy for the diagnosis of OSA during pregnancy [25]. For the analysis domain, five studies did not have reasonable number of participants with the outcome as determined by events per variable (EPV) ≥ 10 for each prediction model (Table 1) [23,24]. All studies selected predictors based on univariate analyses [15,16,17,18,19,20] and only one study properly calibrated model performance by accounting for overfitting, underfitting, and optimism in the model development [17].

### 3.3. Meta-Analysis of Prediction Models

Among 6 studies, there were 29 prediction models, which were classified according to trimesters, development vs. validation phases, and type of pregnancy (high vs. general). A meta-analysis was applied to pool C-statistics of each stratum if there were at least two models.

#### 3.3.1. Development Models

##### Trimester 1

Four studies [15,17,19,20] developed seven prediction models of OSA based on general pregnancy (six models) [17,19,20] and high risk pregnancy (one model) [15]. Pooling overall C-statistics (95% CI) of prediction models with and without high risk pregnancy were 0.817 (0.783, 0850; *I*^2^ = 97.81) and 0.811 (0.768, 0.853; *I*^2^ = 97.58), (see Table 4). Among these models, age and BMI as continuous variables were the common predictors with discrimination C-statistics ranged from 0.772 to 0.800 [19,20]. Model performance improved markedly when frequent snoring, chronic hypertension, and tongue enlargement were included in the models with discrimination C-statistics of >0.80 [15,17,19], whereas models with Sleep Apnea Symptom Score (SASS) or bedpartner reported information did not drastically improve the C-statistics [20].

##### Trimester 2–3

Four studies with 9 prediction models were constructed in the second–third trimesters with the overall pooled C-statistics (95%CI) of 0.855 (0.822, 0.887) with *I*^2^ = 98.06, (see Table 4) [16,17,19,20]. Age and BMI as continuous variables were common predictors in all models with the C-statistics ranged from 0.810 to 0.831 [19,20]. Wilson’s model yielded the highest discrimination C-statistics however it was high ROB from workup and spectrum biases as mentioned while using second trimester data to predict third trimester OSA, and BMI was handled as categorical variable (BMI ≥ 32 kg/m^2^) [16].

#### 3.3.2. External Validation Models

##### Trimester 1

Only one study externally validated 2 prediction models of Multivariable Apnea Prediction Questionnaire (MVAP) [24,26] and Facco models which yielded C-statistics of 0.770 (0.620, 0.920) and 0.800 (0.660, 0.940), respectively (Table 5) [19].

##### Trimester 3

Three studies [16,18,19] externally validated seven prediction models included two MVAP models during third trimester [16,19], 2 MVAP models retrieving data during first or second trimester to predict third trimester OSA [16,19], two Facco models during third trimester [18,19], and 1 Facco model retrieving data during first trimester to predict third trimester OSA, see Table 5 [19]. Discrimination performances C-statistics were 0.643 (0.567, 0.719) to 0.800 (0.670, 0.920) with third trimester pooled C-statistics (95% CI) of 0.736 (0.669, 0.803), *I*^2^ = 96.59, for MVAP models, and 0.752 (0.637, 0.868) to 0.810 (0.710, 0.910) third trimester pooled C-statistics (95% CI) of 0.788 (0.755, 0.821), *I*^2^ = 84.41 for Facco models.

MVAP were externally validated in two studies consisted of five validations in the first and third trimesters, all in general pregnancy with overall pooled C-statistics (95% CI) of 0.743 (0.688, 0.798), *I*^2^ = 95.84. Facco model were externally validated in two studies of high risk and general pregnancy during both first and third trimesters with overall pooled C-statistics (95% CI) of 0.791 (0.767, 0.815), *I*^2^ = 77.34.

#### 3.3.3. Source of Heterogeneity

As shown in Table 4 and Table 5, the results from all meta-analyses showed high heterogeneity (*I*^2^ > 50). Explorative sensitivity analyses were performed according to trimesters. For the first trimester prediction models, the pooled C-statistics (95% CI) for general pregnancy was 0.811 (0.768, 0.853), *I*^2^ = 97.58. The pooled C-statistics (95% CI) were 0.798 (0.761, 0.836), *I*^2^ = 92.56 for PSG and 0.862 (0.842, 0.811, 0.842), *I*^2^ = 83.96 for HSAT studies.

Similar to the third trimester prediction models, the pooled C-statistics (95% CI) were 0.843 (0.826, 0.860), *I*^2^ = 76.39 for PSG and 0.895 (0.783, 1.000), *I*^2^ = 99.74 for HSAT studies.

## 4. Discussion

We conducted a systematic review and meta-analysis of performance of prediction models for OSA during pregnancy. There were 6 studies with 29 eligible prediction models involving 3,713 pregnant women [15,16,17,18,19,20]. Our findings indicated that the existing prediction models showed good performances in discrimination with the pooled C-statistics of 0.817 (0.783, 0.850) and 0.811 (0.768, 0.853) for the first and second–third trimesters OSA, respectively. Two models were externally validated, i.e., MVAP and Facco models yielding fair discrimination performances.

For the developmental viewpoints, all included prediction models demonstrated C-statistic >70, which is considered as threshold for good performance [24,27]. All prediction models included common predictors of age, and BMI as continuous variables, except for Wilson model [16]. When looking at performance of age and BMI as a prediction model from Izci-Balserak, et al. studies, the C-statistics were 0.772–0.800, and 0.831–0.851 for the first and third trimesters, respectively [19,20]. Indicating that BMI and age as continuous variables strongly predicted OSA during pregnancy as described by previous studies [19,28]. In addition, the discriminative performance of the models that included BMI and age were consistent across all trimesters including the new-onset OSA model [17]. This may indicate that BMI and age are the predisposing risk factors for gestational OSA, which can be precipitated by other physiological changes of pregnancy progression. Considering additional predictors such as frequent snoring reports, tongue enlargement into the BMI and age models as Facco, Louis, and BATE models did could increase the discrimination performance C-statistics about 6.25–8.75%; whereas, adding symptom score (SASS), and bed partner reported information did not markedly improve performance of the models [15,17,19,20]. Data from Izci-Balserak studies, using SASS, the symptomatology-based score alone showed less predictive performance with C-statistics of 0.72(0.58, 0.86) and 0.57 (0.43, 0.71) for the first and third trimesters [20]. This is probably because women are less likely to report other symptoms of OSA (apnea, and gasping/choking) [19,21]. However, snoring symptom per se may still prove to be the cardinal symptoms of OSA during pregnancy as by itself demonstrated significant coefficients (i.e., 1.5 in Facco model; 2.4 in Wilson model) [15,16] and were included in many OSA prediction models [15,16,17]. The prevalence of snoring increased significantly during pregnancy particularly those with preeclampsia due to the narrowing of upper airway and were associated with increased adverse maternal and fetal outcomes [29,30,31,32,33,34]. New-onset snoring during pregnancy had also shown association with adverse pregnancy outcome [31,35]. However, only Louis prediction model demonstrated performance for the new-onset OSA [17]. Of notice, none of the prediction models included EDS. As shown in studies, EDS was not discriminative between pregnant women with or without OSA [15,18,36], likely due to the hypersomnolence and sleep disruption of pregnancy itself [21,37]. Other studies had shown that EDS (ESS > 10) was not associated with snoring and gestational hypertension/diabetes [21,29,37]. However, high level of EDS (ESS > 16) was associated with gestational diabetes and other symptoms of OSA (loud snoring, gasping, choking/apnea) [37,38].

Although these prediction models showed good performance in discrimination of OSA in pregnancy, they were high ROBs for many reasons. Majority were lack of calibration and internal validation; low EPV for the construct of the model [15,16,19,20]. Some had workup and spectrum biases and used the predictors in first/second trimesters to predict third trimester OSA [16,19].

Although Facco prediction model yielded lower performance in external setting than the original development setting [15], C-statistics of 0.784 vs. 0.850, it showed good performances for the first and third trimester OSA and both for general and high risk pregnancy [18,19].

As for the MVAP model, the external validations performance was good during first and third trimesters for general pregnancy [16,19]. There was no validation study for high risk pregnancy.

All meta-analyses showed high heterogeneity, sources of heterogeneity were therefore explored by performing sensitivity analyses but no explanation was found. Prediction models constructed based on OSA diagnosed by both PSG and HSAT still showed high discriminative performance but with high heterogeneity. This may be due to numerous variations from different studies (i.e., differences in the clinical setting of the study, type of participants, ethnicity, trimester of screening, diagnostic methods and criteria, prevalence of OSA and the complexity of statistical analysis used to construct each prediction model). However, there were limited prediction model studies, insufficient for further sensitivity analysis. Despite many prediction models had been constructed, only a few models were externally validated, thus generalization of these models is still questionable. This indicates the need for more prediction model studies. Although a few models were externally validated, the findings were also based on high heterogeneity across studies. Caution must be taken to apply prediction models in different clinical settings to that of the original model.

## 5. Conclusions

Evidence from a systematic review and meta-analysis of performance of prediction model for OSA during pregnancy showed good performances during the 1st and 2nd–3rd trimesters. BMI, age, and snoring were the most common predictors among the strong prediction models. The combination of BMI and age included in the models as continuous variables showed consistently good results, and maybe more appropriate across ethnicity. Despite many prediction models developed, only Facco and MVAP models were externally validated. Furthermore, there was high level of heterogeneity indicating the limitation on generalizability and the need for more studies for both development and validation in different clinical settings.

## Figures and Tables

**Figure 1 diagnostics-11-01097-f001:**
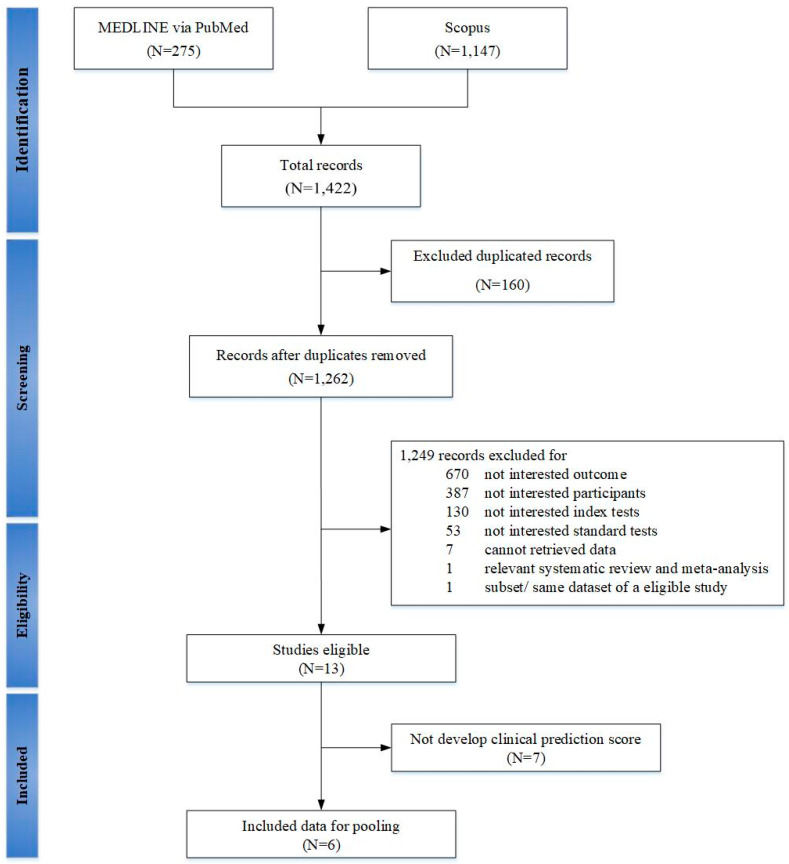
PRISMA flow chart of selecting studies for systematic review and meta-analysis with prediction model in gestational obstructive sleep apnea.

**Figure 2 diagnostics-11-01097-f002:**
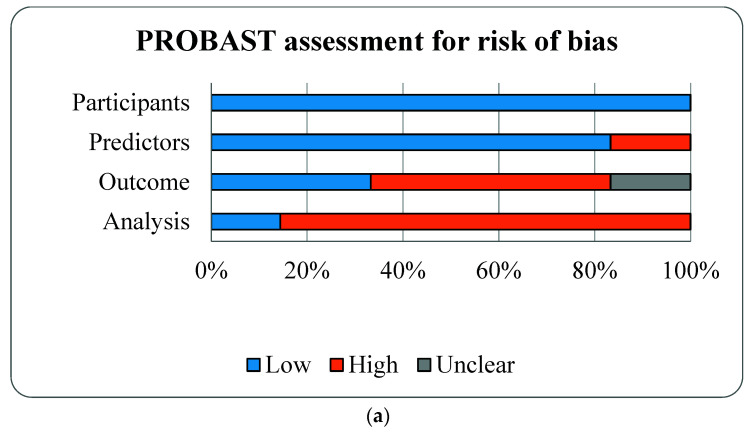
Risk of bias assessment (**a**) and applicability (**b**) using the Prediction model Risk of Bias Assessment Tool (PROBAST) based on four domains of prognostic models for outcome prediction in gestational obstructive sleep apnea.

**Table 1 diagnostics-11-01097-t001:** Characteristics of prediction models for gestational obstructive sleep apnea.

Author, Year and Models	Type of Prediction Study	Study Design	Country	EPV	Ethnicity	Patients
Facco FL. 2012 [15]		Prospective	USA		Mixed	High risk *
First trimester Facco model	Development	cohort		7.0	White 40%,	Nulliparous (27%)
					Black 28%,	and multiparous
					Hispanic 17%,	
					Other 15%	
Wilson LD. 2013 [16]		Prospective	Australia		Data not shown	General
Second trimester Wilson model predicting third	Development	Cohort		5.0		
trimester OSA						
Second trimester MVAP model predicting third	External validation			3.8		
Trimester						
Third trimester MVAP model	External validation			3.8		
Louis JM. 2018 [17]		Prospective	USA		Mixed	General
First trimester model	Development	Cohort		38.3	At recruitment	Nulliparous
Second–third trimester model	Development			69.0	white 60.4%,	(100%)
New-onset OSA model at second–third trimesters	Development			44.0	Black 12.7%,	
					Hispanic 18.3%,	
					Asian 3.8%,	
					others 4.8%	
Dominguez JE. 2018 [18]		Prospective Cohort	USA		Mixed	High risk **
Third trimester Facco model	External validation			4.8	White 29%	
					African	
					American 63.8%	
					Multiple 6.9%	
Izci-Balserak B. 2019 [19]		Cohort	USA		Mixed	General Nulliparous (48.8%) and multiparous
First trimester BATE model	Development			4.3	African	
First trimester Age and BMI model	Development			6.5	American 75%	
First trimester MVAP model	External validation			3.3	White 20%	
First trimester Facco model	External validation			3.3	Other 5%	
First trimester data predict third trimester BATE β Model	Development			7.0		
First trimester data predict third trimester Age and BMI model	Development			10.5		
First trimester data predict third trimester MVAP Model	External validation			5.3		
First trimester data predict third trimester Facco Model	External validation			5.3		
Third trimester BATE model	Development			7.0		
Third trimester Age and BMI	Development			5.3		
Third trimester MVAP	External validation			5.3		
Third trimester Facco	External validation			5.3		
Balserak BI. 2019 [20]		Cohort	USA		Mixed	General Nulliparous (49.2%) and multiparous
First trimester model Age and BMI	Development			8.0	African	
First trimester model SASS combined model I §	Development			5.3	American 75%	
First trimester model SASS combined model II †	Development			4.0	White 20%	
Third trimester model Age and BMI	Development			14.0	Other 5%	
Third trimester model SASS combined model I	Development			9.3		
Third trimester model SASS combined model II	Development			7.0		

Abbreviations: SASS = Sleep Apnea Symptom Score; BMI = Body mass index; MVAP = Multivariable Apnea Prediction Questionnaire; EPV = events per variable, * High risk is defined in the study as those with chronic hypertension (diagnosed prior to pregnancy); obesity (pre-gestational diabetes (type1 or 2); obesity (pre-pregnancy BMI ≥ 30 kg/m^2^), and/or a prior history of preeclampsia. ** High risk is defined in the study as BMI ≥ 40 kg/m^2^ at enrollment; β BATE model = BMI, age, tongue enlargement; § SASS combined model I = SASS + BMI + Age; † SASS combined model II = SASS +BMI + Age+ Bedpartner-reported information.

**Table 2 diagnostics-11-01097-t002:** Characteristics of studies that developed prediction model for gestational obstructive sleep apnea.

Author, Year	Diagnostic Sleep Test/Criteria	Recruitment Trimester (GA)	N	OSA Prevalence	Age, Years Mean (SD)	BMI, kg/m^2^ Mean (SD)	GA, Weeks Mean (SD)	Comments
Facco FL. 2012 [15]	Watch-PAT100 software algorithm; AHI ≥ 5	1 (6–20 weeks)	100	28/100 (28.0%) Median AHI = 1.5 (IQR 0.5, 6.0)	33.0 (6.5)	31.9 (9.1)	16.5 (3.7)	A four-variable screening tool was developed in high risk pregnancy during first trimester using integer-based score from logistic regression (coefficient-based) yielded good sensitivity (86%) and specificity (74%), better than Berlin questionnaire and Epworth sleepiness scale. The model needs further external validation in general and late pregnancy for generalizability.
Wilson DL. 2013 [16]	Polysomnography; AASM 2007 alternative rules of hypopnea; RDI ≥ 5	3 (37 weeks)	43	15/43 (35.0%)	33.5 (5.1)	32.2 (8.0)	22.3 (4.0)	The initial cohort were 380 pregnant women in second trimester, but only subset of participants (43) with either end of sleep-disordered breathing risk (low vs. high risks) underwent PSG during third trimester creating spectrum and workup biases. The new model was constructed based on categorical variables of BMI ≥ 32, snoring volume, and tiredness upon awakening. Model was constructed based on logistic regression (coefficient-based) using second trimester data to predict third trimester OSA. Sensitivity and specificity were 85, and 92%. MAP was externally validated.
Louis JM. 2018 [17]	6-Channel HSAT AASM 2012 hypopnea with ≥3% O_2_ desaturation; AHI ≥ 5	1 (6–15 weeks) 2–3 (22–31 weeks) (New-onset) (22–31 weeks)	3264 2512 2258	1st trimester OSA 115/3264 (3.5%) 3rd trimester OSA 207/2512 (8.2%) New-onset OSA 132/2258 (5.8%)	26.8 (5.6) 27.0 (5.4) 27.0 (5.4)	26.4 (6.4) 26.4 (6.3) 29.2 (6.1)	6–15 22–31 22–31	Predictors were selected by group’s consensus and *p* < 0.15 with forward selection using logistic regression (odd ratio-based). Transformed BMI was used due to skewness. Models were internally validated with 10-fold cross-validation to reduce model optimism. Cutoff ≥ 0.082 yielded sensitivity and specificity of 61 and 90% for first trimester. Cutoff ≥ 0.170 yielded sensitivity and specificity of 54.5 and 90% for third trimester. Cutoff ≥ 0.082 yielded sensitivity and specificity of 45.8 and 90% for new-onset OSA.
Dominguez JE. 2018 [18]	HSAT (ApneaLink) software algorithm; AHI ≥ 5	3 (24–35 weeks)	80	19/80 (24.0%)	30.3 (1.8)	49.8 (3.2)	30.1 (1.4)	The study performed external validation on Facco’s prediction model in pregnant women with extreme obesity (BMI ≥ 40 kg/m^2^). C-statistics was 0.752 with sensitivity and specificity of 100 and 0.21, better than Berlin, Stop-bang, and Epworth sleepiness scale.
Izci-Balserak B. 2019 [19]	Polysomnography AHI ≥ 5 R&K hypopnea with 50% airflow reduction with ≥3% O_2_ desaturation or arousal	1 (<14 week) 3 (n/a)	121 87	10.7% 24.1%	27.1 (6.9) 27.3 (7.2)	30.7 (7.4) 33.3 (6.6)	12.2 (2.0) 33.8 (2.4)	A longitudinal cohort of OSA screening during first and third trimester developed BATE prediction model using backward stepwise logistic regression (coefficient-based). The model consisted of BMI, age, and tongue enlargement. The study identified tongue enlargement as significant predictor of OSA during both trimesters, unrelated to race, although 75% of participants were African American. Sensitivity and specificity were 76 and 82% for first and third trimesters.
Balserak BI. 2019 [20]	Polysomnography AHI ≥ 5 R&K hypopnea with 50% airflow reduction with ≥3% O_2_ desaturation or arousal	1 (<14 week) 3 (n/a)	126 105	12.7% 26.7%	27.2 (7.2) 27.2 (7.2)	30.0 (7.1) 33.4 (6.4)	12.0 (1.9) 33.6 (2.5)	Using the same cohort of healthy pregnant women to construct two combined prediction models to Sleep Apnea Symptom Score (SASS). Model I had sensitivity and specificity of 77 and 74% for first trimester; 77 and 78% for third trimester. Model II had sensitivity and specificity of 77 and 72% for first trimester; 82 and 78% for third trimester.

Abbreviations: SASS-Sleep Apnea Symptom Score; BMI-Body mass index; OSA-obstructive sleep apnea; HSAT-home sleep apnea test; ASSM-American Academy of Sleep Medicine; R&K-Rechtschaffen and Kales; AHI-apnea-hypopnea index; RDI-respiratory disturbance index; GA-gestational age; standard deviation-standard deviation.

**Table 3 diagnostics-11-01097-t003:** Tabular Presentation for The Prediction model Risk of Bias Assessment Tool (PROBAST) Results of studies of gestational obstructive sleep apnea.

Study	ROB				Applicability			Overall	
	Participants	Predictors	Outcome	Analysis	Participants	Predictors	Outcome	ROB	Applicability
Facco FL. 2012 [15]	+	+	+	-	+	+	+	-	+
Wilson LD. 2013 [16]	+	-	-	-	+	+	+	-	+
Louis JM. 2018 [17]	+	+	-	+	+	+	+	-	+
Dominguez JE. 2018 [18]	+	+	-	-	+	+	+	-	+
Izci-Balserak B. 2019 [19]	+	+	+	-	+	+	+	-	+
Balserak BI. 2019 [20]	+	+	?	-	+	+	+	-	+

Abbreviations: ROB = Risk of bias, + Low risk; - High risk; ? Unclear.

**Table 4 diagnostics-11-01097-t004:** Descriptions and meta-analyses of discriminative performances of prediction model in developmental phase for gestational obstructive sleep apnea.

Trimester	Author, Year	Patients	Model	Predictors	Development C-Statistics (95%CI)	Internal Validation C-Statistics (95%CI)
1	Facco FL. 2012 [15]	High risk *	Facco model	Age, BMI †, Frequent snoring ᵟ, Chronic HT	0.850 (0.770, 0.930)	-
	Louis JM. 2018 [17]	General	Louis model	Age, Transformed BMI §, Frequent snoring ᵟ	0.870 (0.848, 0.891)	0.870 (0.848, 0.891)
	Izci-Balserak B. 2019 [19]	General	BATE model	Age, BMI †, Tongue enlargement	0.860 (0.760, 0.960)	-
		General	Age and BMI model	Age, BMI †	0.800 (0.650, 0.940)	-
	Balserak BI. 2019 [20]	General	Age and BMI model	Age, BMI †	0.772 (0.635, 0.909)	-
		General	SASS combined model I	Age, BMI †, SASS	0.776 (0.639, 0.914)	-
		General	SASS combined model II	Age, BMI †, SASS, Bed partner reports γ	0.781 (0.648, 0.914)	-
	Pooled C-statistics of all 7 models in the first trimester *Q-test = 212.93, p-value < 0.001, I^2^* = *97.81*				0.817 (0.783, 0.850)	-
	Pooled C-statistics of 6 models in the first trimester included only general pregnancy. *Q-test = 206.91, p-value < 0.001, I^2^* = *97.58*				0.811 (0.768, 0.853)	-
3	Wilson LD. 2013 [16]	General	Wilson model	BMI ≥ 32 kg/m^2^, Snoring volume, Tired upon awaking	0.952 (0.914, 0.989)	-
	Louis JM. 2018 [17]	General	Louis model	Age, Transformed BMI §, Frequent snoring ᵟ	0.838 (0.821, 0.855)	0.838 (0.821, 0.855)
	Izci-Balserak B. 2019 [19]	General	BATE model	Age, BMI †, Tongue enlargement	0.870 (0.770, 0.960)	-
		General	Age and BMI model	Age, BMI †	0.810 (0.710, 0.920)	-
		General	BATE model (using first trimester data)	Age, BMI †, Tongue enlargement	0.870 (0.770, 0.970)	-
		General	Age and BMI model (using first trimester data)	Age, BMI †	0.850 (0.750, 0.940)	-
	Balserak BI. 2019 [20]	General	Age and BMI model	Age, BMI †	0.831 (0.714, 0.947)	-
		General	SASS combined model I	Age, BMI †, SASS	0.826 (0.708, 0.943)	-
		General	SASS combined model II	Age, BMI †, SASS, Bed partner reports γ	0.842 (0.732, 0.952)	-
	Pooled C-statistics of 9 all models in the third trimester *Q-test = 412.25, p-value < 0.001, I^2 =^ 98.06*				0.855 (0.822, 0.887)	-
3 New-onset OSA	Louis JM. 2018 [17]	General	New-onset OSA model	Age, Transformed BMI §, Frequent snoring ᵟ	0.809 (0.786, 0.832)	0.809 (0.786, 0.832)

Abbreviations: SASS-Sleep Apnea Symptom Score; HT-hypertension; BMI-Body mass index; CI-Confidence interval, * High risk is defined in the study as those with chronic hypertension (diagnosed prior to pregnancy); obesity (pre-gestational diabetes (type 1 or 2); obesity (pre-pregnancy BMI ≥ 30 kg/m^2^), and/or a prior history of preeclampsia, ᵟ Frequent snoring is defined as self-reported snoring ≥ 3 times/week, † Age and BMI used as continuous variables, § Transformed BMI is defined as (BMI^λ^ − 1)/g, where (g = geometric mean BMI ^(λ − 1)^), as continuous variable. Tongue enlargement is defined if tongue protrudes beyond the teeth or the alveolar ridge in the resting position, γ Bed partner reports were obtained if participants had bedpartner by questions asking the frequency of loud snoring and long pauses between breath while asleep during the past month.

**Table 5 diagnostics-11-01097-t005:** Description and meta-analyses of discriminative model performances for external validation of MVAP and Facco models on gestational obstructive sleep apnea.

Trimester	Author, Year	Patients	Model	Predictors	External Validation C-Statistics (95%CI)
1	Izci-Balserak B. 2019 [19]	General	MVAP model	Age, BMI †, Sex, SASS	0.770 (0.620, 0.920)
3	Izci-Balserak B. 2019 [19]	General	MVAP model		0.770 (0.640, 0.890)
		General	MVAP model (using first trimester data)		0.800 (0.670, 0.920)
	Wilson LD. 2013 [16]	General	MVAP model (using seond trimester data)		0.733 (0.660, 0.806)
		General	MVAP model		0.643 (0.567, 0.719)
Pooled C-statistics of all 5 MVAP models *Q-test = 96.17, p-value < 0.001, I^2^* = *95.84*					0.743 (0.688, 0.798)
Pooled C-statistics of 4 MVAP models including only third trimester *Q-test = 87.88, p-value < 0.001, I^2^* = *96.59*					0.736 (0.669, 0.803)
1	Izci-Balserak B. 2019 [19]	General	Facco model	Age, BMI †, Frequent snoring **, Chronic HT	0.800 (0.660, 0.940)
3	Izci-Balserak B. 2019 [19]	General	Facco model		0.800 (0.700, 0.910)
		General	Facco model (using first trimester data)		0.810 (0.710, 0.910)
	Dominguez JE. 2018 [18]	High risk *	Facco model		0.752 (0.637, 0.868)
Pooled C-statistics of all 4 Facco models *Q-test = 13.24, *p*-value = 0.004, I^2^* = *77.34*					0.791 (0.767, 0.815)
Pooled C-statistics of 3 Facco models including only third trimester *Q-test = 12.83, p-value = 0.002, I^2^* = *84.41*					0.788 (0.755, 0.821)

Abbreviations: SASS-Sleep Apnea Symptom Score; MVAP-Multivariable Apnea Prediction; HT-hypertension, HT; BMI-Body mass index; CI-Confidence interval, * High risk is defined in the study as BMI ≥ 40 kg/m^2^ at enrollment. ** Frequent snoring is defined as self-reported snoring ≥ 3 times/week. † Age and BMI used as continuous variables.

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
