# Peer review of "Prediction Models of Obstructive Sleep Apnea in Pregnancy: A Systematic Review and Meta-Analysis of Model Performance"

_diagnostics, 2021, doi:10.3390/diagnostics11061097_

Round 1

Reviewer 1 Report

I have read the article by Siriyotha et al. with great interest. The authors investigated models to predict OSA in pregnancy. The systematic review and meta-analysis is well conducted and the article is well written.

Comments:

  • Inclusion and exclusion of studies. Why only medline and scopus? What about other sources? This needs to be discussed.
  • What happened if there was a disagreement between the two authors who selected the studies? Who decided on the outcome?
  • In Tables, why are there journal names in brackets for ref 19 and 20 and not for the others?
  • Watch-PAT is not (yet) an endorsed diagnostic tool for OSA. Please, discuss this limitation.

Author Response

Response to reviewers

Thank you very much for your valuable suggestions and comments. We would like to response to your comments as below.

Comments from Reviewer1:

  1. Inclusion and exclusion of studies. Why only medline and scopus? What about other sources? This needs to be discussed.

            Response

  • According to the PRISMA recommendation, at least 1 databases should be used. The Medline and Scopus were among the major ones, we add the sentences in line 71-72. But we agreed that the more the better.

“An extensive literature search was performed on 2 major databases in MEDLINE (from 1996 to 28 February, 2021), and Scopus databases (from 1980 to 28 February, 2021) as recommended by the PRISMA guideline.”

  1. What happened if there was a disagreement between the two authors who selected the studies? Who decided on the outcome?

Response

  • Two independent authors selected the study (SS, and SR), if there is a disagreement between the 2 authors another author (VT) decided on the outcomes. We added the sentence on line 92-93

“Disagreements between reviewers were adjudicated by a 3rd reviewer (VT).”

  1. In Tables, why are there journal names in brackets for ref 19 and 20 and not for the others?

Response

  • Because the 2 articles were from the same author and the same year. But as the reviewer suggested, we decided to delete the journal names since they were already referenced and to make the same format with other references.
  1. Watch-PAT is not (yet) an endorsed diagnostic tool for OSA. Please, discuss this limitation.

Response

  • We agree that Watch-PAT is not the gold standard diagnostic tool for OSA, it is still polysomnography. But for watch-PAT, it had been studied and validated for the diagnosis of OSA during pregnancy. So, we added the sentences in line 178-180.

“Although PSG is the gold standard diagnostic test for OSA, only 3 studies performed PSG. Three      studies used HSAT with different criteria for hypopneas [15, 17, 18]. One study used Watch-PAT, a wrist-worn device using a peripheral arterial tonometry (PAT), finger plethysmography, and pulse oximeter. Watch-PAT had been validated and shown good accuracy for the diagnosis of OSA during pregnancy.”

Reviewer 2 Report

Siriyotha et al. performed a systematic review and meta-analysis to evaluate multivariable prediction models both development and validation involving diagnosis of OSA during pregnancy. This study followed the PRISMA and PICO guidelines. Of 1,262 articles, only 6 studies (patient number =3,713) met the inclusion criteria and were included for review. They found that all studies showed a high risk of bias for the construct of models. The pooled C-statistics (95%CI) for development prediction models was 0.817 (0.783, 0850) and 0.855 (0.822, 0.887) for the 1st and 2nd-3rd trimesters, respectively. Only MVAP and Facco models were externally validated with lower pooled C-statistics (95%CI) of 0.743 (0.688, 0.798) and 0.791 (0.767, 0.815). The body mass index, age, and snoring were the most common predictors in the models. They concluded that prediction models for gestational OSA showed good performance during early and late trimesters but a high level of heterogeneity limits for generalizability. Generally, this study well prepared and well written. It highlights the potential application of reliable clinical prediction models to screen gestational OSA. I recommend considering acceptance of this manuscript after minor revisions.
1. Some typos and grammar issues can be corrected after careful editing.
2. Please specify "n = ?" in the abstract and main text.
3. Please provide detailed descriptions of sleep apnea symptom frequency index (SASS), frequent snoring, tongue enlargement, and bed partner reports because they determined by different operational definitions.
4. Can you obtain all available patient data and externally verify these interesting prediction models?

Author Response

Response to reviewers

Thank you very much for your valuable suggestions and comments. We would like to response to your comments as below.

Comments from Reviewer2:

  1. Some typos and grammar issues can be corrected after careful editing.

Response

  • We carefully edit some typos and grammar issue as the reviewer suggested. We edit the title of the study to “Prediction models of obstructive sleep apnea in pregnancy: A systematic review and meta-analysis of model performance”. This corresponded to the title registered to the PROSPERO.
  1. Please specify "n = ?" in the abstract and main text.

Response

  • We edited and specify the numbers as suggested by the reviewer in the abstract and the main text on line 18, 132, 134, 144, 145
  1. Please provide detailed descriptions of sleep apnea symptom frequency index (SASS), frequent snoring, tongue enlargement, and bed partner reports because they determined by different operational definitions.

Response

- Detailed description of all the prediction models including the SASS were provided in the appendix, frequent snoring, tongue enlargement, and bed partner reports according to different reference were provided in the supplementary appendix B line 362-405 as indicated in line 102, 103. Additional information for frequent snoring and tongue enlargement, and bed partner reports were added into the abbreviation of Table 4.

  1. Can you obtain all available patient data and externally verify these interesting prediction models?

Response

- No, we only performed systematic review and meta-analysis. Data were extracted from what the author reported. We did not obtain individual patient data.

Reviewer 3 Report

Dr Siriyotha and coworkers present a systematic review and meta-analysis regarding the prediction models of obstructive sleep apnea in pregnancy. The topic is important and there is a need for a better understanding of mechanisms and symptoms as well as complications of gestational obstructive sleep apnea. In this context, the current manuscript is a good contribution to the field. However, the authors should provide some additional information regarding:

1) The analyzes presented rely on an AHI cut-off 5 for the OSA diagnosis but the specificity and sensitivity analyzes can be improved if AHI cut-off 15 is also added.

2) A sensitivity analysis including only PSG studies is necessary because the lack information on total sleep time from the HSAT data may confound the results. 

3) Change of the title to: "Prediction models of obstructive sleep apnea in pregnancy: a systematic review and meta-analysis of model performances" "women" should be deleted. Ä°ts should be either "pregnant women" or "pregnancy".

Author Response

Response to reviewer 3

Thank you very much for your valuable comments and suggestions. We have made corrections according to your comments below

1) The analyzes presented rely on an AHI cut-off 5 for the OSA diagnosis but the specificity and sensitivity analyzes can be improved if AHI cut-off 15 is also added.

    Response: It would be very interesting to include the analysis if AHI cut-off of 15 is used. But unfortunately, we were unable to perform sensitivity analysis for this. Because none of the included studies had perform diagnosis with AHI cut-off at 15 events/hour. Majority used AHI≥5, and 1 study used RDI≥5 events/hour, (see table 2 for characteristics of the included studies).

2) A sensitivity analysis including only PSG studies is necessary because the lack information on total sleep time from the HSAT data may confound the results. 

    Response: We included the sensitivity analysis of PSG studies and HSAT study in line 239-246 and in Discussion section in line 326-329 (highlighted in yellow in the manuscript) below:

“Source of heterogeneity

As shown in tables 4-5, the results from all meta-analyses showed high heterogeneity (I2>50). Explorative sensitivity analyses were performed according to trimesters. For the 1st trimester prediction models, the pooled C-statistics (95%CI) for general pregnancy was 0.811 (0.768, 0.853), I2=97.58. The pooled C-statistics (95%CI) were 0.798 (0.761, 0.836), I2=92.56 for PSG and 0.862 (0.842, 0.811, 0.842), I2= 83.96 for HSAT studies.   

Similar to the 3rd trimester prediction models, the pooled C-statistics (95%CI) were 0.843 (0.826, 0.860), I2=76.39 for PSG and 0.895 (0.783, 1.000), I2= 99.74 for HSAT studies.”

“All meta-analyses showed high heterogeneity, sources of heterogeneity were therefore explored by performing sensitivity analyses but no explanation was found. Prediction models constructed based on OSA diagnosed by both PSG and HSAT still showed high discriminative performance but with high heterogeneity.”

3) Change of the title to: "Prediction models of obstructive sleep apnea in pregnancy: a systematic review and meta-analysis of model performances" "women" should be deleted. Ä°ts should be either "pregnant women" or "pregnancy".

      Response

    We changed the title to “Prediction models of obstructive sleep apnea in pregnancy: a systematic review and meta-analysis of model performances” as suggested.

Some additional minor corrections to the references were made (highlighted in green in the manuscript).

Thank you very much again for your valuable time to review our manuscript

Visasiri Tantrakul, MD

Round 2

Reviewer 3 Report

I have no additional comments. Congratulations.

Author Response

Dear Editor,

Thank you very much for your valuable suggestions and comments.

  1. Line 336-338: Please clarify these sentences. “Although a few models were externally validated, the findings were based on high heterogeneity across studies, generalizability of these model maybe problematic.”
  • Response: these sentences had been modified below in 331-332:

“Although a few models were externally validated, the findings were also based on high heterogeneity across studies. Caution must be taken to apply prediction models in different clinical settings to that of the original model.”

  1. “Abbreviations” in these footnotes. For the abbreviations first write the abbreviation, acronym, “=”, then full name/s. Table 2. I believe it will be clearer if “Trimester GA” (title of the 3rd column) is replaced with “recruitment time”, “recruitment trimester” or “recruitment GA”
    Tables in Appendix A: Please replace TABLE 1 with TABLE S1 and TABLE 2 with TABLE S2.
    There are a couple of sentences starting with “But” please replace “but” with “However” and there are couple of places.  
  • Response: Corrections were made as suggested (highlighted in yellow) on lines 300, 327, 370.
